# Investigating Chaperone like Activity of Green Silver Nanoparticles: Possible Implications in Drug Development

**DOI:** 10.3390/molecules27030944

**Published:** 2022-01-29

**Authors:** Mohd Ahmar Rauf, Md Tauqir Alam, Mohd Ishtikhar, Nemat Ali, Adel Alghamdi, Abdullah F. AlAsmari

**Affiliations:** 1Department of Pharmaceutical Sciences, Wayne State University, Detroit, MI 48201, USA; mxr2481@miami.edu; 2Department of Biochemistry, Aligarh Muslim University, Aligarh 202002, Uttar Pradesh, India; 3Department of Biochemistry and Biophysics, Texas A&M University, College Station, TX 77843, USA; ishtikharm_20@tamu.edu; 4Department of Pharmacology and Toxicology, College of Pharmacy, King Saud University, P.O. Box 55760, Riyadh 11451, Saudi Arabia; nali1@ksu.edu.sa (N.A.); 441105941@student.ksu.edu.sa (A.A.)

**Keywords:** human serum albumin (HSA), aggregates, nanotechnology, B-AgNPs, circular dichroism (CD), thioflavin T (ThT), congo red (CR)

## Abstract

Protein aggregation and amyloidogenesis have been associated with several neurodegenerative disorders like Alzheimer’s, Parkinson’s etc. Unfortunately, there are still no proper drugs and no effective treatment available. Due to the unique properties of noble metallic nanoparticles, they have been used in diverse fields of biomedicine like drug designing, drug delivery, tumour targeting, bio-sensing, tissue engineering etc. Small-sized silver nanoparticles have been reported to have anti-biotic, anti-cancer and anti-viral activities apart from their cytotoxic effects. The current study was carried out in a carefully designed in-vitro to observe the anti-amyloidogenic and inhibitory effects of biologically synthesized green silver nanoparticles (B-AgNPs) on human serum albumin (HSA) aggregation taken as a model protein. We have used different biophysical assays like thioflavin T (ThT), 8-Anilino-1-naphthalene-sulphonic acid (ANS), Far-UV CD etc. to analyze protein aggregation and aggregation inhibition in vitro. It has been observed that the synthesized fluorescent B-AgNPs showed inhibitory effects on protein aggregation in a concentration-dependent manner reaching a plateau, after which the effect of aggregation inhibition was significantly declined. We also observed meaningful chaperone-like aggregation-inhibition activities of as-synthesized florescent B-AgNPs in astrocytes.

## 1. Introduction

Several neurodegenerative chronic disorders like Alzheimer’s, Parkinson’s, systemic amyloidosis, etc., involve the aggregation and deposition of misfolded proteins ranging from small oligomers to large amyloid masses [1,2]. Amyloids are insoluble proteins characterized by the highly ordered β-sheet rich structural motif [3]. Also, to date, none of the drugs has been developed which can halt amyloidosis completely since the process is relatively slow and takes many years to be diagnosed. Amyloidosis follows the unfolding mediated misfolding/aggregation of native proteins [4]. Proteins are functional in their native conformations, which are formed through controlled folding of their fundamental primary structure in a sequential fashion [5]. Apart from amino acid residues present in the primary polypeptides, other environmental factors may play important roles in gaining functional native structure of proteins [6]. Unfolding-mediated aggregation of native proteins is a critical stage in amyloidosis that is influenced by a variety of environmental factors [7]. Most of the protein aggregates are found to be rich in cross-β sheets which facilitate larger surface area for amyloidosis to take place [8]. Elucidating the insights of protein unfolding mediated aggregation pathways will help in designing better drugs for amyloids related disorders.

Nowadays, nanotechnology is gaining the focus of major scientific communities due to its broad applicability. It is a multidisciplinary scientific field that is witnessing rapid growth. Despite its enormous scope, one of nanotechnology’s most significant benefits will be the development of innovative and successful medical treatments. Understanding the interactions of nanomaterials and biomolecules both in extracellular and inside the cells will open up new strategies for developing effective nanomedicines [9,10,11]. Nanomaterials are synthesized and modified accordingly nowadays for the gain of a variety of functions. Previously it has been reported that the reduced-cysteine residues have strong interactions with silver nanoparticles (AgNPs), which play roles in maintaining its native conformations [12]. Biologically synthesized (B-AgNPs) interact with proteins and form a dynamic B-AgNPs-protein corona which probably influences unfolding and refolding pathway resulting in protein aggregation inhibition and chaperone-like activity [13]. But the formation of protein corona may affect the properties of AgNPs and the conformation and function of itself, which may lead to toxicity [14]. Also, it has been shown that protein-nanoparticle interactions are primarily electrostatic; therefore, surface charge on both of them may play essential roles in their stability and functions [15]. Hence it becomes essential to probe the nature of interactions between proteins and NPs under various conditions.

In the present study, we first characterized the bio-synthesized green AgNPs, (referred as B-AgNPs) using *Rosa* petals extracts by spectroscopy, dynamic light scattering (DLS), and microscopy. Then, we elucidated the mechanism through which B-AgNPs prevent protein aggregation in a concentration-dependent manner in vitro. The role of biogenic AgNPs in inhibiting aggregation was investigated by carefully controlling the artificial environment on the aggregation pathway of human serum albumin (HSA) using several assays such as circular dichroism (CD), ANS fluorescence, Thioflavin T (ThT) fluorescenceCongo red (CR) assay, and turbidity assay. Additionally, we examined the uptake of these B-AgNPs in astrocytes to determine their ability to suppress protein aggregation under in vitro conditions.

## 2. Materials and Methods

### 2.1. Materials

The Appendix A contains all of the materials utilized in this research. The procedures are given below:

#### 2.1.1. Biological Synthesis of B-AgNPs by *Rosa* (Rose) Petals

B-AgNPs were synthesized via reducing the silver precursor, silver nitrate, mediated by *Rosa* petal extract (AgNO_3_). The synthesis was carried out by incubating a fixed volume of silver nitrate (1 mM AgNO_3_) with various quantities of the OS stock (150–500 µL) [16,17]. Deionized water was used to adjust the final volume of the reaction mixture to 5 mL. To achieve B-AgNPs, the reaction mixture was continuously agitated at room temperature for the specified time duration. The sample was centrifuged at 10,000× *g* for 10 min to remove the unwanted plant products/materials. Finally, the pellet was lyophilized to obtain the nanoparticles. The synthesized B-AgNPs were then characterized employing spectroscopy, Transmission electron microscope (TEM) [TEM, H-7500, Hitachi Ltd., Tokyo, Japan and Dynamic light scattering [(DLS (Beckman Coulter Delsa Nano particle size analyzer (Miami, FL, USA)] techniques for their shape and size analysis.

#### 2.1.2. Characterization of B-AgNPs

##### UV-Visible Spectroscopy

The UV-Vis absorption spectra of silver ions were examined to determine their biogenic reduction to colloidal nanostructures (B-AgNPs) [18]. UV-Visible scanning [Hitachi, Tokyo, Japan] of the incubation mixture at different time points was used to determine the kinetics of B-AgNPs production.

##### Transmission Electron Microscopy (TEM) and Dynamic Light Scattering (DLS)

The hydrodynamic particle size was studied in Beckman Coulter Delsa Nano particle size analyzer (Miami, FL, USA) [19] at ambient temperatures. The morphology of the nanoparticles was analyzed by transmission electron microscopy (TEM, H-7500, Hitachi Ltd., Tokyo, Japan) [20]. The detailed information has been described in the Appendix A.

#### 2.1.3. Protein Sample Preparation

Stock solution (5 mg/mL) of HSA was prepared in a sodium phosphate buffer of pH 7.4 and then left overnight for dialysis. Lowry’s assay was used for determining the concentration of protein stock solution [21,22]. Detailed information for the sample preparation has been described in the Appendix A.

#### 2.1.4. In Vitro Exploration of the Role of B-AgNPs on the Aggregation Pathway of HSA

##### Turbidity Assay

The preliminary conformational change analysis in the native HSA in 5 M urea and increasing concentrations of B-AgNPs was done by performing a turbidity assay [23]. The methodology for the turbidity analysis have been described in the Appendix A.

##### 8-Anilino-1-Naphthalene-Sulphonic Acid (ANS) Fluorescence Measurements

ANS (molecular formula C_16_H_13_NO_3_S) is a dye that binds to the hydrophobic regions of a polypeptide [24,25], is employed to study the change in the conformation of the protein. The details methodology have been described in the Appendix A.

##### Far-UV Circular Dichroism Analysis

Changes in the secondary structure of HSA was analyzed first by Far-UV CD spectral analysis [26]. The detailed protocol has been described in the Appendix A.

##### Thioflavin T Fluorescence Assay

The binding of Thioflavin T (ThT) dye with β-amyloid gives characteristic spectra which has been used for unfolding mediated aggregation of proteins for many years. ThT (molecular formula C_17_H_19_ClN_2_S) has been used for decades as the gold standard for detecting amyloid fibrils in solution and tissue secretions. Spectro fluorophotometer (Shimadzu RF-5301, Kyoto, Japan) was used for the ThT spectral analysis. Quartz cuvette of path length 1 cm was used. Excitation was done at 440 nm and emission spectra were recorded at 450–600 nm. Double distilled water (pH 7.0) was used for the preparation of ThT solution. The final concentration of HSA used was 5 μΜ while that of ThT was 20 μΜ. Also, fluorescence microscopy was used to visualize the fluorescence associated with ThT-bound HSA amyloid. ThT possesses excitation and emission spectrum peaks at around 450 and 490 nm, respectively, responsible for the observed green fluorescence. Briefly, the amyloids from different groups were taken and plated on a coverslip glass on a glass slide and observed under Flourescence microscopy (FM) for the FITC channel (Alam et al., 2018b) [27,28], In addition, Interaction of various amyloid species obtained at various time points with ThTwas also assessed by fluorescence microscopy. Incubated HSA aggregates obtained at various time points, were pelleted down at 15,000× *g* for 15 min. The pellets obtained were incubated with 20 µM ThT for 30 min at room temperature and then transferred onto a glass slide to be analyzed under fluorescence microscope and images were observed at 10X magnification (Olympus, Tokyo, Japan). [29,30].

##### Congo Red Assay

Congo red (molecular formula C_32_H_22_N_6_Na_2_O_6_S_2_) is a dye that binds to β-amyloid of aggregated proteins and causes a redshift in the spectra as compared to the native proteins. UV-Vis1700, spectrophotometer (Tokyo, Japan), was used for taking CR absorbance measurements in the range of 400–700 nm using a quartz cuvette of path length 1 cm. Like ThT analysis, the CR based imaging was also performed for the different groups and slides were observed under FM for the red fluorescence channel (TriTC) at 10X magnification [31,32].

##### Transmission Electron Microscopy

Finally, TEM analysis was done to visualize the native and incubated samples of HSA in the presence of different increasing concentrations of B-AgNPs [33,34]. The detailed methodology for the TEM protocol have been described in the Appendix A.

#### 2.1.5. Uptake of B-AgNPs by Astrocyte

Astrocytes (1 × 10^6^) were plated on a coverslip on a 6 well plate at 37 °C in a humidified atmosphere at 5% CO_2_ were treated with variable B-AgNPs concentration for 2 h. After 2 h of incubation, the cells were washed with sterile PBS (pH 7.4) to remove excess AgNPs. After washing, the astrocytes were observed in a fluorescent microscope (Olympus, Tokyo, Japan) at 10X to check their uptake.

#### 2.1.6. Assay for Cell Viability Using MTT Assay

The MTT (3-(4,5-dimethylthiazol-2-yl)-2,5-diphenyltetrazolium bromide) assay was used to investigate the toxicity of HSA aggregate on astrocyte cells. In 96-well plates, cells were grown in DMEM-F-12 (10% FBS) medium until they reached 70% confluency. Cells were rinsed with sterile PBS and then treated for 24 h at 37 °C in 5% CO_2_ with early-stage HSA oligomers produced in the presence/absence of B-AgNPs. The detailed protocol has been described in the Appendix A.

#### 2.1.7. Statistical Analysis

All of the tests were carried out in triplicate and the results were expressed as the standard error mean. Graph Pad Prism 7.0 was used to analyze the data after the one-way analysis of variance (California, USA). The statistically significant difference at *p* < 0.05 was considered [35].

## 3. Results and Discussion

### 3.1. Biological Synthesis of B-AgNPs by Rosa (Rose) Petals (Characterization of B-AgNPs)

In our studies, we used *Rosa* flower extract to synthesize silver nanoparticles. *Rosa Santana* is a prominent ornamental Rosaceae cultivar. The flower is rich in tannin and pectin, as well as fatty acids and organic acids [36]. Additionally, rose water, which is created by steeping petals in water, contains a plethora of potent antioxidants and is widely utilized as a fragrance in the food and cosmetic industries. Dried flower petals are consumed as herbal tea to treat digestive tract illnesses [37]. Due to these qualities; this plant acts as an effective reducing agent, facilitating the creation of silver nanoparticles from silver ions.

The synthesis of silver nanoparticles in aqueous petal extract was investigated using absorption spectra spanning 300–700 nm wavelength. The shift in colour of the silver nitrate solution to blackish-brown with the addition of petals extract was interpreted as evidence for the creation of silver nanoparticles. At 432 nm, a single high signal was found in the spectra, confirming the production of the silver nanoparticles (Figure 1A). The UV–Vis spectroscopy results corroborated prior studies indicating that silver nanoparticles have a surface plasmon resonance (SPR) peak between 410 and 450 nm due to their spherical shape [38]. The transmission electron microscopy revealed that the as-synthesized nanoparticles were composed of well-dispersed NPs of variable shape and size. The majority of the particles as-synthesized were spherical to oval in form with a diameter of 10 to 50 nm (Figure 1B). Figure 1C,D demonstrate the zeta potential and particle size distribution of silver nanoparticles. Silver nanoparticles synthesized are monodisperse, with an average diameter of 42 nm, according to the particle size distribution. The average zeta potential of silver nanoparticles was found to be −16.30 mV. The strong negative potential value improves long-term stability and colloidal nature of silver nanoparticles due to negative–negative charge repulsion [39].

### 3.2. Analyzing Changes in the Native Conformation of HSA

We initially examined the structural changes in native HSA and HSA incubated with increasing concentrations of B-AgNPs in the presence of 5 M urea using turbidity measurements at 360 nm. The study aimed to analyze the effect of B-AgNPs on the aggregation of HSA induced by urea at physiological temperature, i.e., 37 °C, in critically designed in vitro conditions. Our previous study found that the B-AgNPs (size 10–50 nm) inhibited protein aggregation induced by urea maximally at 30 μg/mL (*w*/*v*) when the protein samples were incubated at a temperature mimicking the pathological temperature during fever, i.e., 40 °C in vitro. In the current study, we discovered that B-AgNPs inhibited protein aggregation in a concentration-dependent manner. Still, the maximum inhibition was observed at 35 μg/mL (*w*/*v*), above which the effect of protein aggregation inhibition was reduced or became insignificant, as indicated by turbidity assay measurements (Figure 2A). The turbidity of the sample containing native HSA was found to be the lowest, while that of the sample containing HSA in the presence of 5 M urea was found to be the highest, as expected. Turbidity was shown to diminish in HSA and urea samples incubated with increasing doses of B-AgNPs in a concentration-dependent manner, with the maximum inhibition appearing at 35 μg/mL (*w*/*v*) B-AgNPs. Above 35 μg/mL (*w*/*v*) B-AgNPs, the incubated samples showed slightly increased turbidity, which might be due to decreased inhibition of protein aggregation induced by urea above that concentration of B-AgNPs. ANS fluorescence measurements further analyzed changes in the native conformations and aggregation inhibition in HSA by B-AgNPs. It is well established that ANS binds with the hydrophobic residues of the polypeptides, which is exposed during the unfolding of a native protein. ANS spectra also supported our findings that B-AgNPs inhibited protein aggregation in a concentration-dependent manner reaching its maximum at 35 μg/mL (*w*/*v*) B-AgNPs at 37 °C. Figure 2B shows that the sample containing native HSA has minimum ANS fluorescence intensity, which is usual since, in the native conformation of a protein, hydrophobic residues are buried inside when present in aqueous polar solutions [40]. The highest ANS fluorescence intensity was observed when HSA was treated with 5 M urea, possibly due to unfolding-mediated aggregation of HSA, exposing hydrophobic residues. HSA and 5 M urea samples incubated with increasing doses of B-AgNPs demonstrated a decrease in ANS fluorescence intensity up to 35 µg/mL (*w*/*v*) B-AgNPs, at which point the fluorescence intensity increased. This demonstrated that the highest inhibition of protein aggregation occurred at a concentration of 35 µg/mL (*w*/*v*) of B-AgNPs, above which the effect of protein aggregation inhibition was diminished, perhaps due to B-AgNPs self-aggregation. However, when we incubated the HSA with varying concentrations of B-AgNPs in the absence of urea, the nanoparticles doesn’t showed prominent changes in the native conformations of HSA and aggregation is not observed as was observed in the presence of 5 M urea [Appendix A].

### 3.3. Analyzing Changes in the Secondary Structure of HSA

Circular dichroism (CD) spectroscopy is a frequently used technique for quantifying the secondary structure and unfolding of proteins [41,42]. The recommended assay is based on the ability of a CD spectrometer to measure not just differential absorption between right and left circularly polarized light, but also normal absorption, or more precisely, turbidity. This is accomplished by registering the dynode voltage V, which is a high voltage applied to the UV detector’s photomultiplier to compensate for light intensity losses due to absorption and scattering [43].

Far-UV CD spectral measurements were used to investigate changes in the secondary structure of native HSA. Far-UV CD spectral analysis provides information about a protein’s secondary structure changes [44]. The native HSA exhibited two fold minima at 208 and 222 nm, typical feature of an alpha-helical protein (Figure 3) [45]. The HSA sample incubated with 5 M urea revealed a single negative minimum at around 217 nm, indicative of the dominant cross-sheet structure reported in protein aggregates [46]. At around 217 nm, the abrupt decrease in negative ellipticity suggested the production of HSA aggregates in the presence of 5 M urea. The incubation of HSA with urea in the presence of increasing concentrations of B-AgNPs resulted in an increase in negative ellipticity approaching to that of native HSA, which could result from B-AgNPsinduced aggregation inhibition. The effect of B-AgNPs on protein aggregation was maximal at 35 µg/mL (*w*/*v*), above which no appreciable aggregation inhibition was seen. Although this is not a confirming test for protein aggregation and inhibition of protein aggregation, but it provides the most plausible explanation for the phenomena mentioned above, which was confirmed by many experiments.

### 3.4. Confirming Aggregate Formation and Aggregation Inhibition in HSA

The dye thioflavin T (ThT) is widely regarded as the gold standard for detecting protein aggregates [47]. It has been utilized to identify aggregates in vitro and various tissues, including cerebrospinal fluids, by seeding the disease-related protein recombinant with patient fluid [48]. ThT forms complexes with amyloid aggregates [49]. In this assay, it was observed that native HSA exhibited the lowest ThT fluorescence intensity, whereas after incubating HSA with 5 M urea exhibited the highest ThT fluorescence intensity (Figure 4A), probably due to urea-induced unfolding-mediated aggregation. It was observed that when HSA samples were incubated with 5 M urea in the presence of increasing quantities of B-AgNPs, the ThT fluorescence intensity decreased, likely due to B-AgNPs inhibiting protein aggregation. Again, the effect of protein aggregation inhibition was most significant in the presence of 35 µg/mL (*w*/*v*) B-AgNPs, above which the effect was considerably reduced. Similar to ANS assay, we performed ThT assay, where we incubated the HSA in the presence of increasing concentrations of B-AgNPs without 5 M urea. The ThT and ANS spectral analysis in the absence of 5 M urea doesn’t show noticeable impact on the native conformation of protein, also aggregation is not noticed as was observed with 5 M Urea [Appendix A].

In concordance with ThT fluorescence spectral studies, fluorescence microscopy also revealed a similar pattern, i.e., aggregates obtained after urea incubation exhibited increased fluorescence. The B-AgNPs chaperone activity was observed in Fluorescence Microscope under FITC channel shows that amyloidal aggregate production was modest (nearly nonexistent) when HSA was co-incubated with B-AgNPs at 35 μg/mL (*w*/*v*) compared to untreated, which shows all green-fluorescent aggregates (Figure 4B). For further evaluation for the effect of protein aggregation inhibition induced by B-AgNPs, CR assay was performed. It is well known that CR also binds to β-amyloids of aggregated proteins and gives a redshift in CR absorbance spectra with enhanced absorbance intensity. In this study, the native HSA gave minimum absorbance with absorbance maxima at around 490 nm. The incubated sample of HSA in the presence of 5 M urea showed a sharp increase in CR absorbance with absorbance maxima at around 500 nm (Figure 5A). The enhanced absorbance with red shift of 10 nm is clear indication of HSA aggregation induced by urea. HSA with 5 M urea in the presence of increasing concentrations of B-AgNPs showed decrease in CR absorbance reaching to native HSA which could only be due to protein aggregation inhibition induced by B-AgNPs. The effect of protein aggregation inhibition was maximum in the presence of 35 μg/mL (*w*/*v*) B-AgNPs, above which there was significant decrease in the protein aggregation inhibition by B-AgNPs. Like, ThT based microscopic analyses we also carried out image analysis for the B-AgNPs based inhibition for amyloidal aggregate formation after B-AgNPs incubation for CR based dye. The data observed under fluorescence microscopy reciprocates the results observed under spectroscopy, suggesting its rationality as a chaperone agent (Figure 5B).

### 3.5. Morphological Analysis of Aggregates

Finally, TEM examination was used to demonstrate the effect of B-AgNPs on protein aggregation inhibition. Figure 6A illustrates that the incubated sample containing native HSA does not exhibit any aggregated condition, whereas the sample containing 5 M urea displays protein aggregates (Figure 6B). Additionally, the incubated sample containing 35 µg/mL (*w*/*v*) of B-AgNPs (Figure 6C) exhibited fewer aggregated states as a result of the inhibition of protein aggregation, but the effect of protein aggregation inhibition is reduced above 35 µg/mL (*w*/*v*), as demonstrated here with 55 µg/mL (*w*/*v*) of B-AgNPs (Figure 6D).

### 3.6. Uptake of B-AgNPs by Astrocytes

When fluorescent active B-AgNPs were incubated with astrocytes, the cells took them up. The intrinsic red fluorescence of B-AgNPs indicates that they have been internalized. Once within the cells, it is expected that the B-AgNPs will exert their chaperone ability (Figure 7A).

### 3.7. B-AgNPs Protects Astrocyte from HSA-Amyloidal Aggregates Mediated Cytotoxicity

To determine the amount of HSA-amyloidal aggregate cytotoxicity, the MTT assay was used. The HSA amyloidal aggregates treatment leads to around 62% viability in the cells. Suggesting, HSA amyloidal aggregates have cellular toxicity. In contrast, the cell viability was enhanced to 70% and 82% respectively, in the presence of 5 μg/mL and 35 μg/mL (*w*/*v*) B-AgNPs treated amyloidal aggregates. This finding confirms that B-AgNPs’ anti-amyloidogenic activity increased cell viability. ANOVA was utilized to analyze the MTT results (Figure 7B). We hypothesized that the presence of B-AgNPs would diminish or lessen amyloidal aggregates formation. Non-soluble amyloidal aggregates produced will cause cellular toxicity in live cells. B-AgNPs aid to delay the overall amyloid synthesis process by extending or increasing the lag phase of the amyloid production process by targeting early aggregation species. To put it another way, hazardous aggregates were captured by B-AgNPs. Since, the intermediates are no longer available once trapped, the synthesis of amyloid got inhibited or minimised [50,51].

The depletion of hazardous aggregates by B-AgNPs results in a reduction in astrocytes toxicity. Additionally, incubation of B-AgNPs with astrocytes resulted in their uptake into the cells, as evidenced by the presence of fluorescing B-AgNPs within the cells (Figure 7A). Once within, B-AgNPs may act as a chaperone, preventing protein misfolding in astrocytes. The MTT data clearly indicate that the B-AgNPs have antiaggregant properties. The nanoparticles in a dose-dependent manner block the formation of harmful aggregates, hence regulating the mortality of the treated cells.

## 4. Conclusions

The present work demonstrated the effect of different concentrations of B-AgNPs, on the putative conformational changes in HSA. AgNPs have seen a dramatic surge in application in the pharmaceutical and food industries due to their long-known multifunctional properties. That is why it is critical to examine the effect of nanoparticles on macromolecules’ shape and stability. We believe that the synthesized smaller size B-AgNPs may be beneficial in suppressing protein aggregation up to a concentration of roughly 35 µg/mL. Above this point, the inhibitory impact diminishes. As a result, the use of smaller nanoparticles requires additional in vivo testing and should be carefully evaluated. The study is a preliminary examination to determine the influence of B-AgNPs on the inhibition of aggregation of HSA in the presence of urea. One could argue that the prevailing conditions in natural systems are significantly more complicated, with a diversity of proteins and other factors interacting with the protein under study. However, the current study can serve as a springboard for future efforts to establish an effective technique for preventing the protein-associated amyloid formation and unravel metal nanoparticles’ role as a potent inhibitor of amyloid/fibril formation. This study will aid in the design of in vivo experiments to determine the effect of B-AgNPs on animal models, which will aid in the development of more effective drugs for aggregation-related disorders and will also serve as a guide for the pharmaceutical and food industries in terms of limiting their use of AgNPs. Our findings may pave the way for nanoparticle-based therapies for disorders associated with protein aggregation.

## Figures and Tables

**Figure 1 molecules-27-00944-f001:**
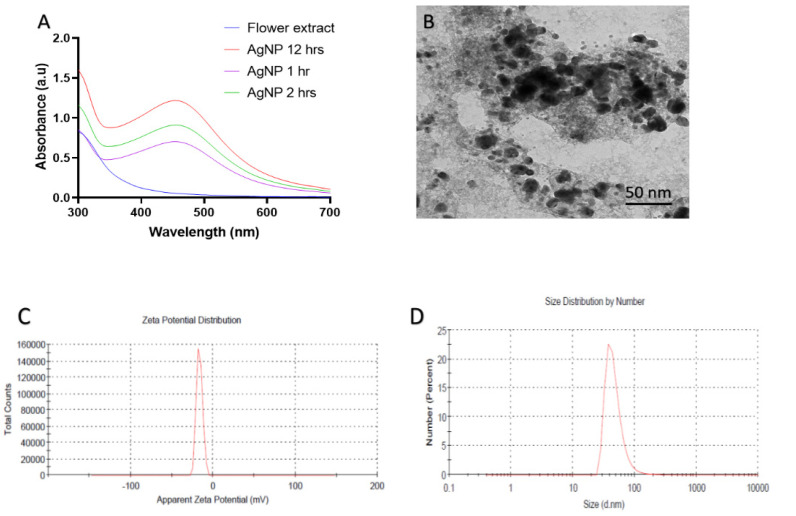
**Physical characterization of the synthesized B-AgNPs.** (**A**) UV–visible absorption spectra of B-AgNPs synthesized using *Rosa* petal extract: UV-Vis spectra of silver nanoparticles generated when AgNO_3_ (1 mM) was incubated with *Rosa Santana* petal extract. As the petal extract are added to the incubation mixture, the distinctive surface plasmon resonance (SPR) bands corresponding to silver nanoparticles gradually shift toward longer wavelengths with associated band intensity amplification. (**B**) TEM study of as-synthesized B-AgNPs reveals their shape and size: A representative TEM picture of silver nanoparticles generated from *Rosa Santana* petal extract. TEM micrograph demonstrating the coexistence of spherical and oval nanostructures of B-AgNPs formed during the incubation of *Rosa Santana* petal extract in aqueous AgNO_3_. (**C**) Zeta potential of the synthesized B-AgNPs. (**D**) Size study of the B-AgNPs using DLS (Dynamic Light Scattering): indicates that the as-synthesized silver nanoparticles have an overall particle radius of around 10–50 nm.

**Figure 2 molecules-27-00944-f002:**
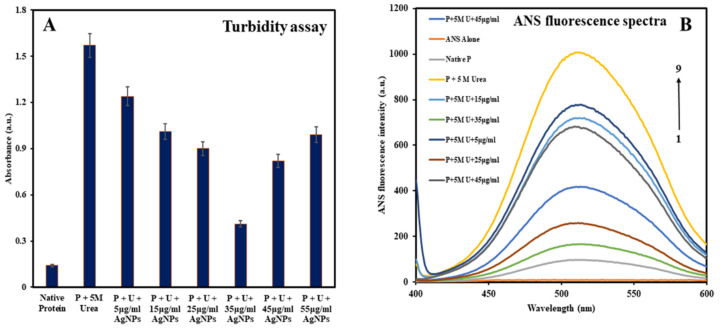
(**A**) Turbidity assay of native HSA, HSA containing 5 M urea, and HSA containing 5 M urea in the presence of increasing quantities of B-AgNPs. Each experiment was repeated three times. The final protein concentration was 5 µM, and the turbidity was determined at 360 nm. (**B**) ANS fluorescence spectra of native HSA, HSA incubated with 5 M urea, and HSA incubated with increasing concentrations of B-AgNPs in addition to the fixed concentration of 5 M urea. The excitation wavelength was 380 nm, and the emission wavelength was between 400 and 600 nm. The width of the excitation slit was set to 10 nm, while the width of the emission slit was set to 5 nm.

**Figure 3 molecules-27-00944-f003:**
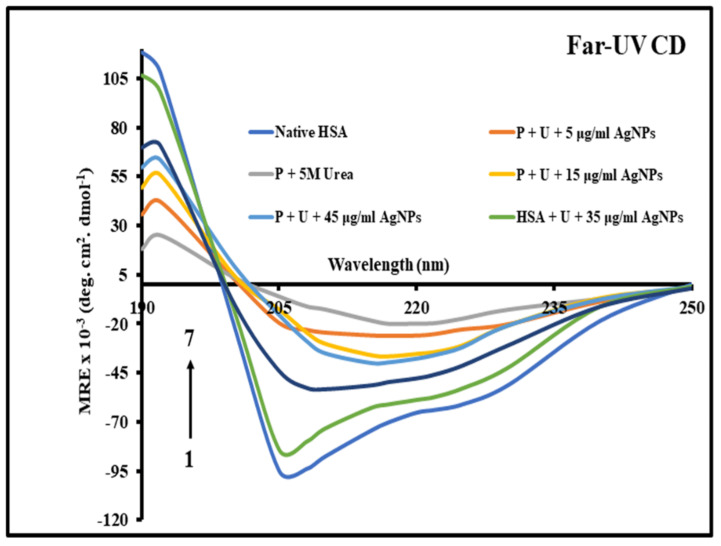
Far UltraViolet CD spectra of native HSA, HSA in the presence of 5 M urea, and HSA in the presence of increasing concentrations of B-AgNPs. Between 190 and 250 nm, the spectra were recorded. The final concentration of protein was 5 µM, and the path length was 0.1 cm.

**Figure 4 molecules-27-00944-f004:**
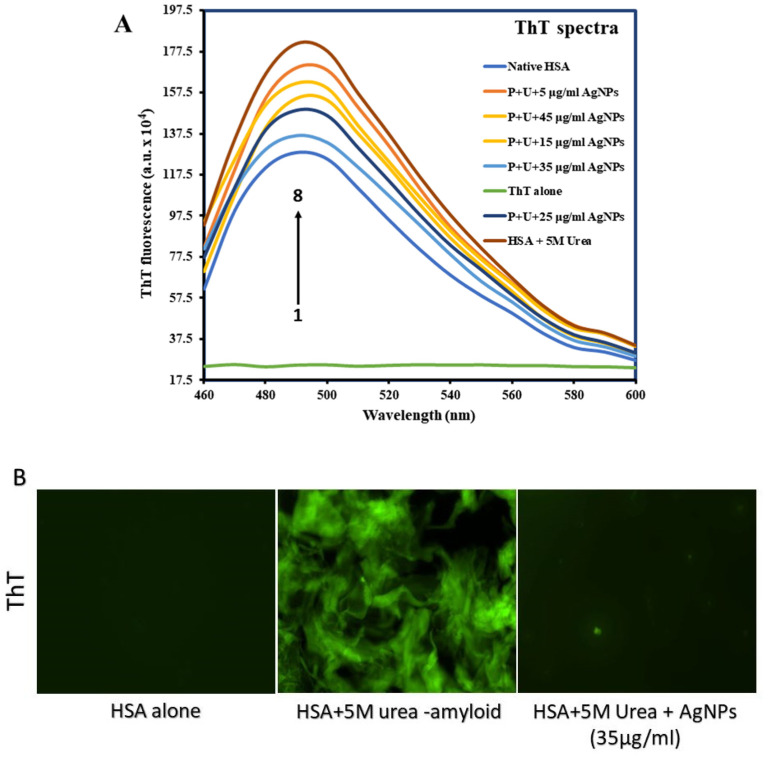
(**A**) ThT fluorescence spectra of native HSA (curve 2), HSA in the presence of 5 M urea (curve 8) and HSA in the presence of increasing concentrations of B-AgNPs (curves 3–7). HSA was used at a final concentration of 5 µM, while ThT was used at a final concentration of 20 µM. (**B**) Green color fluorescence indicates HSA amyloidal aggregates coupled to ThT. Fluorescence microscopy is used to visualize HSA amyloidal aggregates. Micrographs of HSA aggregates as generated in the presence or absence of B-AgNPs at 10X magnification at 100 µm scale.

**Figure 5 molecules-27-00944-f005:**
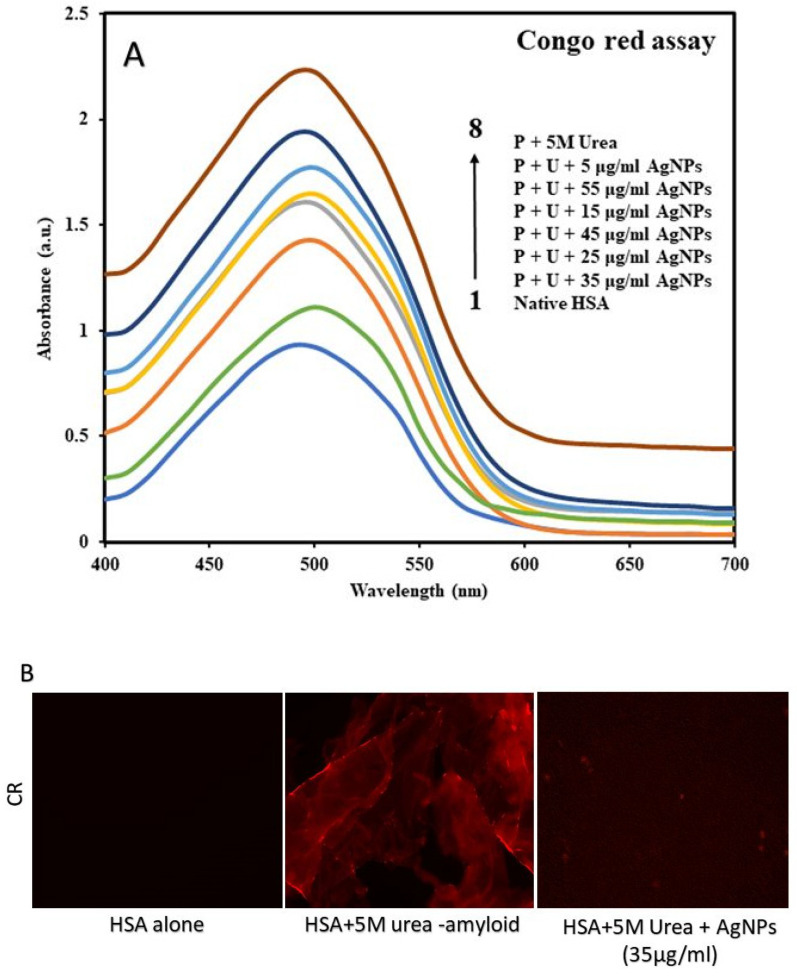
(**A**) CR absorption spectra of native HSA, HSA incubated with 5 M urea, and HSA incubated with increasing doses of B-AgNPs. The final concentration of protein was 0.8 µM, and the pathway length was 1 cm. (**B**) Red fluorescence correlates to HSA-amyloidal aggregate coupled to the CR dye: Micrographs of HSA amyloidal aggregates generated in the presence or absence of B-AgNPs at 10X magnification at 100 µm scale.

**Figure 6 molecules-27-00944-f006:**
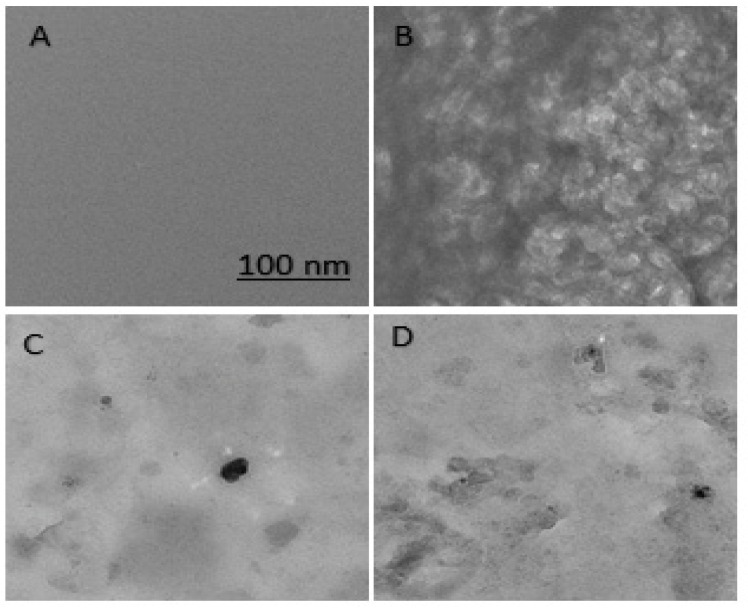
Transmission electron microscopy analysis of incubated samples for native HSA, HSA with 5 M urea, and HSA with 5 M urea in the presence of 35 µg/mL (*w*/*v*) B-AgNPs and 55 µg/mL (*w*/*v*) (**A**–**D**) respectively. The image is observed at 20,000X magnification at scale bar of 100 nm.

**Figure 7 molecules-27-00944-f007:**
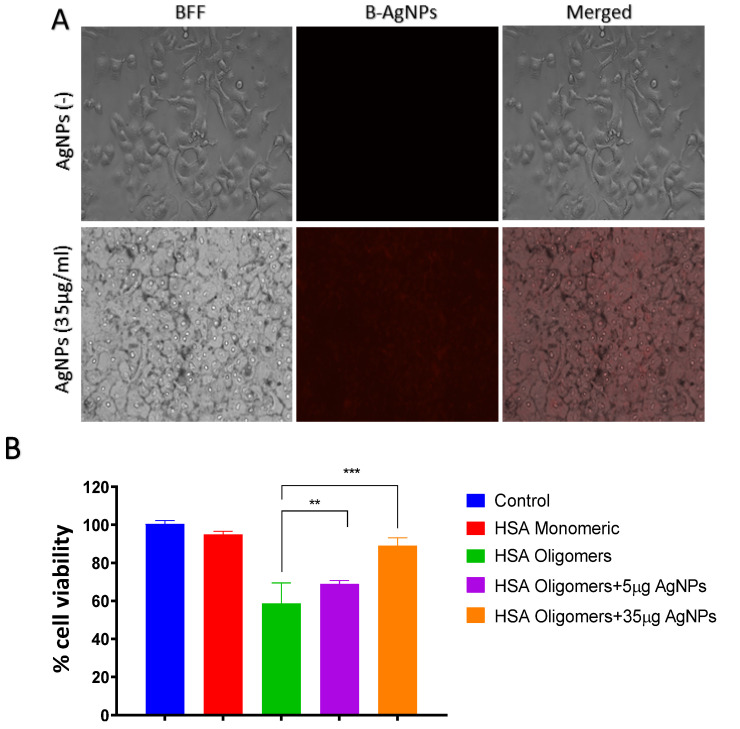
(**A**) **Astrocytes uptake B-AgNPs:** Astrocytes were grown overnight on the glass cover slip. When B-AgNPs were incubated with astrocytes, the NPs were taken up in a concentration-dependent manner. Micrographs of cells in the absence or presence of B-AgNPs at a magnification scale of 10X. (**B**) **MTT assay for Cell Viability:** The MTT cell viability assay was used to determine the capacity of as-synthesized B-AgNPs to suppress the formation of hazardous HSA aggregates. Although HSA aggregates are toxic to the cells, the addition of B-AgNPs inhibits the formation of HSA aggregates. Experiments were performed in triplicates, results are shown as mean ± SD; ** *p* ≤ 0.01; *** *p* ≤ 0.001.

## Data Availability

Data is available within the article and Appendix A.

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
