# Peer review of "Investigating Chaperone like Activity of Green Silver Nanoparticles: Possible Implications in Drug Development"

_molecules, 2022, doi:10.3390/molecules27030944_

Round 1
Reviewer 1 Report
- In figure 2.B, the description of peaks is not readable. please change the font color blue to black.
- Why do you use urea in the experiments and what its basis?
- One experiment should be done without urea i.e. (HSA + AgNPs).
- Add the discussion and cite the suitable references in support and contrast of your findings from page no. 8 to last.
- Insert the suggested references and do some suggested changes.

Author Response
Reviewer 1: In figure 2.B, the description of peaks is not readable. please change the font color blue to black.
Response: Font color has been changed from blue to black.
Why do you use urea in the experiments and what its basis?
Response: Urea has been reported to cause protein aggregation above 3M. It has been taken as a protein aggregating agent to create and analyze a critically designed in vitro environment for aggregation inhibition studies.
One experiment should be done without urea i.e. (HSA + AgNPs).
Response: We have performed thioflavin t assay and ANS fluorescence assay in the absence of urea and submitted as supplementary data.
Add the discussion and cite the suitable references in support and contrast of your findings from page no. 8 to last.
Response: Thank you reviewer for your valuable comments, we have inserted the suggested changes in the revised version.
Insert the suggested references and do some suggested changes.
Response: Thank you reviewer for your valuable comments, we have inserted the suggested references
Reviewer 2:
In Abstract, please describe briefly the methodology used and some of the mostrelevant results, and reduce the part represented by literature data;
Response: Thank you reviewer for your valuable comments, we have modified the said comment in the abstract.
Please explain all the used abbreviations at their first use and include them all into the Abbreviations section;
Response: Abbreviations have been added.
In Methods (both in the main text and SI), please provide the complete origin of the used substances, devices and apparatuses, or software (company, city, (state) and country);
Response: The comment has been resolved.
In Methods in SI – in TEM, please explain why the AgNPs were stained with 1% uranylacetate (or perhaps it would be better to remove this statement);
Response: Thank you reviewer for your valuable comments, we have modified the said comment in the SI file.
Reviewer 3:
In the abstract section, the authors have written ‘’till now….’’ Please change it, it looks haphazard.
Response: Thank you reviewer for your valuable comments, we have modified the said comment in the abstract.
The authors should have included line numbers, so that it would be easy for thereviewers to look for mistakes, like in the synthesis section, the authors have written150–500 L OS stock, is it litre, microliters or millilitres?
Response: Thank you reviewer for your valuable comments, we have modified the said comment.
The authors should include the point why they think there NPs are fluorescent?
Response: We have performed the cell uptake assay to show the fluorescent nature of AgNPs (Figure 7A).
The author should include the ThT, Congo red and CD methodology in the main file,rather than the SI file.
Response: We have modified as per suggesttion.
Why after reaching a concentration of 35μg/ml, it starts showing plateau and impactstarts going down for inhibitory action of AgNPs?
Response: Inserted comment in the main file
Figure 6, is not clear, I expect authors to provide a high-resolution image for properunderstanding.
Response: We have modified the article file and inserted the modified image
In Figures 4B, 5B and 7, please provide the scale bar and write the magnification scale.
Response: Modified as per suggestion.
Check the references with journals guidelines
Response:Modified as per suggestion.
In Results, please reorganize the figures or increase the size of labels on axes of graphical representations, and/or into the legends included into the graphs;
Response: Thank you reviewer for your valuable comments, we have modified the figures as per suggestions.
In figures 4 and 5 it is not clear what are [A] and [B]. please explain into the legend or add labels on the figures’ panels.
Response: Thank you reviewer for your valuable comments, we have modified the figures as per suggestions.

Reviewer 2 Report
Journal: Molecules
Manuscript ID: molecules-1558138
Type of manuscript: Article
Title: Investigating Chaperone like Activity of Green Silver Nanoparticles: Possible Implications in Drug Development
Authors: Mohd Ahmar Rauf, Md Tauqir Alam *, Mohd Ishtikhar, Nemat Ali, Adel Alghamdi, Abdullah F. AlAsmari *
In this manuscript submitted as Article to Molecules journal, the authors (with some experience into this field) reported very interesting results of a well designed and conducted study on the ability of green synthesized AgNPs to reduce protein aggregation. The topic of this manuscript is relevant to the field of the Molecules and fits with the scope of this journal. The paper has a good structure, is well written (apart from a few grammar or typing errors) and concise, and is based on relevant and mostly recent references. The methodology is described in enough detail, and the results are well presented (but their presentation could be improved). The conclusions are based on the obtained results. However, before recommending the publication of this manuscript, this reviewer considers that a minor revision is required. Thus, there are 8 minor aspects to be solved:
- In Abstract, please describe briefly the methodology used and some of the most relevant results, and reduce the part represented by literature data;
- Please explain all the used abbreviations at their first use and include them all into the Abbreviations section;
- In Methods (both in the main text and SI), please provide the complete origin of the used substances, devices and apparatuses, or software (company, city, (state) and country);
- In Methods in SI – in TEM, please explain why the AgNPs were stained with 1% uranyl acetate (or perhaps it would be better to remove this statement);
- In Results, please reorganize the figures or increase the size of labels on axes of graphical representations, and/or into the legends included into the graphs;
- In figures 4 and 5 it is not clear what are [A] and [B]. please explain into the legend or add labels on the figures’ panels;
- In panel A of Figure 6 the reader cannot see anything. Please explain. Also in Figure 6 the scale bar have to be visible (or mention into the legend: “scale bar=XXX nm);
- Some examples of grammar or typing errors:
- p.1 in Abstract: please reformulate “This study was carried out a carefully designed in-vitro analysis”;
- p.3 in 2.1.4.6: please reformulate “The detailed protocol have been described in the SI file”;
- p.4 in 2.1.7: please reformulate “results were expressed as mean standard error of the mean”;
- p.4 in 3.1 1st paragraph: please write rosa with capital (for genus) or rose (as popular name) and use italics for names of family, genus and species (Rosa, Rosa santana, Rosaceae);
- p.4 in 3.1 1st paragraph: please replace “;” with “,”after “qualities”;
- p.5 legend of Figure 1: please reformulate “As the petal extract are added to...”;
- p.6 in 3.3: please reformulate “The recommended assay is based on the The ability of a CD spectrometer”;
- p.2 in SI: in the paragraph for TEM, please replace “transfer electron microscopy” with “transmission electron microscopy”;
- p.4 in SI: in the paragraph for TEM, please use the correct name of the microscope which is JEOL, not JOEL.
Author Response
Reviewer 2:
- In Abstract, please describe briefly the methodology used and some of the mostrelevant results, and reduce the part represented by literature data; Response: Thank you reviewer for your valuable comments, we have modified the said comment in the abstract.
- Please explain all the used abbreviations at their first use and include them all into the Abbreviations section; Response: Abbreviations have been added.
- In Methods (both in the main text and SI), please provide the complete origin of the used substances, devices and apparatuses, or software (company, city, (state) and country); Response: The comment has been resolved.
- In Methods in SI – in TEM, please explain why the AgNPs were stained with 1% uranylacetate (or perhaps it would be better to remove this statement); Response: Thank you reviewer for your valuable comments, we have modified the said comment in the SI file.
- In Results, please reorganize the figures or increase the size of labels on axes of graphical representations, and/or into the legends included into the graphs;Response: Thank you reviewer for your valuable comments, we have modified the figures as per suggestions.
- In figures 4 and 5 it is not clear what are [A] and [B]. please explain into the legend or add labels on the figures’ panels.Response: Thank you reviewer for your valuable comments, we have modified the figures as per suggestions.
Reviewer 3 Report
The manuscript has shown the effect of variable concentrations of silver nanoparticles in a simple lucid. I do have some comments on the article as follows.
- In the abstract section, the authors have written ‘’till now….’’ Please change it, it looks haphazard.
- The authors should have included line numbers, so that it would be easy for the reviewers to look for mistakes, like in the synthesis section, the authors have written 150–500 L OS stock, is it litre, microliters or millilitres?
- The authors should include the point why they think there NPs are fluorescent?
- The author should include the ThT, Congo red and CD methodology in the main file, rather than the SI file.
- Why after reaching a concentration of 35µg/ml, it starts showing plateau and impact starts going down for inhibitory action of AgNPs?
- Figure 6, is not clear, I expect authors to provide a high-resolution image for proper understanding.
- In Figures 4B, 5B and 7, please provide the scale bar and write the magnification scale.
- Check the references with journals guidelines
Author Response
We have modified the manuscript as per the suggestion of learned reviewer.
